# ROBUST POLICY OPTIMIZATION IN DEEP REINFORCEMENT LEARNING

## ABSTRACT

Entropy can play an essential role in policy optimization by selecting the stochastic policy, which eventually helps better explore the environment in reinforcement learning (RL). A proper balance between exploration and exploitation is challenging and might depend on the particular RL task. However, the stochasticity often reduces as the training progresses; thus, the policy becomes less exploratory. Therefore, in many cases, the policy can converge to sub-optimal due to a lack of representative data during training. Moreover, this issue can even be severe in high-dimensional environments. This paper investigates whether keeping a certain entropy threshold throughout training can help better policy learning. In particular, we propose an algorithm Robust Policy Optimization (RPO), which leverages a perturbed Gaussian distribution to encourage high-entropy actions. We evaluated our methods on various continuous control tasks from DeepMind Control, OpenAI Gym, Pybullet, and IsaacGym. We observed that in many settings, RPO increases the policy entropy early in training and then maintains a certain level of entropy throughout the training period. Eventually, our agent RPO shows consistently improved performance compared to PPO and other techniques such as data augmentation and entropy regularization. Furthermore, in several settings, our method stays robust in performance, while other baseline mechanisms fail to improve and even worsen the performance.

## 1 INTRODUCTION

Exploration in a high-dimensional environment is challenging due to the online nature of the task. In a reinforcement learning (RL) setup, the agent is responsible for collecting high-quality data. The agent has to decide on taking action which maximizes future return. In deep reinforcement learning, the policy and value functions are often represented as neural networks due to their flexibility in representing complex functions with continuous action space. If explored well, the learned policy will more likely lead to better data collection and, thus, better policy. However, in high-dimensional observation space, the possible trajectories are larger; thus, having representative data is challenging. Moreover, it has been observed that deep RL exhibit the primacy bias, where the agent has the tendency to rely heavily on the earlier interaction and might ignore helpful interaction at the later part of the training (Nikishin et al., 2022).

Maintaining stochasticity in policy is considered beneficial, as it can encourage exploration (Mnih et al., 2016; Ahmed et al., 2019). Entropy is the randomness of the actions, which is expected to go down as the training progress, and thus the policy becomes less stochastic. However, lack of stochasticity might hamper the exploration, especially in the large dimensional environment (high state and action spaces), as the policy can prematurely converge to a suboptimal policy. This scenario might result in low-quality data for agent training. In this paper, we are interested in observing the effect when we maintain a certain level of entropy throughout the training and thus encourage exploration.

We focus on a policy gradient-based approach with continuous action spaces. A common practice (Schulman et al., 2017; 2015) is to represent continuous action as the Gaussian distribution and learn the parameters ($\mu$, and $\sigma$) conditioned on the state. The policy can be represented as a neural network, and it takes the state as input and outputs the one Gaussian parameters per action dimension. Then the final action is chosen as a sample from this distribution. This process inherently introduces

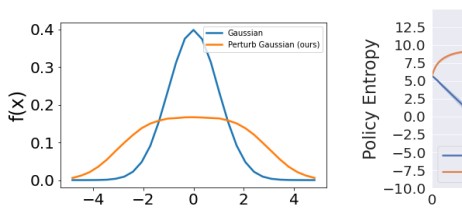 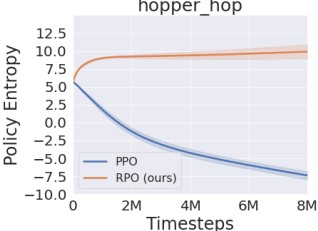 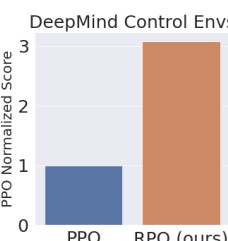

Figure 1: [**Left**] Standard Gaussian and corresponding Perturb Gaussian distribution (ours). We observe that the probability density is less centered around the mean in our distribution. [**Middle**] Policy entropy at different timesteps of training in DeepMind Hopper Hop Environment. Similar patterns are observed for other evaluated environments. The PPO agent who uses Standard Gaussian starts from a particular entropy and becomes less stochastic as the training progresses and reduces the policy entropy. In contrast, our agent RPO uses our perturbed Gaussian distribution, increases the entropy at the initial timestep, and then maintains a certain level of entropy throughout the training. [**Right**] Our agent RPO shows over 3x performance improvement in normalized return compared to the base PPO agent on 12 DeepMind control environments. The results are averaged over 10 random seed runs.

randomness in action as every time it samples action for the same state, the action value might differ. Though, in expectation, the action value is the same as the mean of the distribution, this process introduces some randomness in the learning process. However, as time progresses, the randomness might reduce, and the policy becomes less stochastic.

We first notice that when this method is used to train a PPO-based agent, the entropy of the policy starts to decline as the training progresses, as in Figure 1, even when the return performance is not good. Then we pose the question; what if the agent keeps the policy stochastic or entropy throughout the training? The goal is to enable the agent to keep exploring even when it achieves a certain level of learning. This process might help, especially in high-dimensional environments where the state and action spaces often remain unexplored. We developed an algorithm called Robust Policy Optimization (RPO), which maintains stochasticity throughout the training. We notice a consistent improvement in the performance of our method in many continuous control environments compared to standard PPO.

Seeing the data augmentation through the lens of entropy, we observe that empirically, it can help the policy achieve a higher entropy than without data augmentation. However, this process often requires prior knowledge about the environments and a preprocessing step of the agent experience. Moreover, such methods might result in an uncontrolled increase in action entropy, eventually hampering the return performance (Raileanu et al., 2020; Rahman & Xue, 2022). Another way to control entropy is to use an entropy regularizer (Mnih et al., 2016; Ahmed et al., 2019), which often shows beneficial effects. However, it has been observed that increasing entropy in such a way has little effect in specific environments (Andrychowicz et al., 2020). These results show difficulty in setting proper entropy throughout the agent's training.

To this end, in this paper, we propose a mechanism for maintaining entropy throughout the training. We propose to use a new distribution to represent continuous action instead of standard Gaussian. The policy network output is still the Gaussian distribution's mean and standard deviation in our setup. However, we add a random perturbation on the mean before taking an action sample. In particular, we add a random value $z \sim U(-\alpha, \alpha)$ to the mean $\mu$ to get a perturbed mean $\mu' = \mu + z$. Finally, the action is taken from the perturbed Gaussian distribution $a \sim N(\mu', \sigma)$. The resulting distribution is shown in Figure 1. We see that the resulting distribution becomes flatter than the standard Gaussian. Thus the sample spread more around the center of the mean than standard Gaussian, whose samples are more concentrated toward means. The uniform random number does not depend on states and policy parameters and thus can help the agent to maintain a certain level of stochasticity throughout the training process. We name our approach Robust Policy Optimization (RPO) and compare it with the standard PPO algorithm and other entropy-controlled methods such as data augmentation and entropy regularization.

We evaluated our method in several environments from DeepMind Control (Tunyasuvunakool et al., 2020), OpenAI Gym (Brockman et al., 2016), PyBullet (Coumans & Bai, 2016–2021), and Nvidia IsaacGym (Makoviychuk et al., 2021). We observed that our method RPO performs consistently better than PPO, and the performance improvement shows larger in high-dimensional environments. Moreover, RPO outperform two data augmentation-based method: RAD (Laskin et al., 2020) and DRAC (Raileanu et al., 2020). In addition to that, our method is simple and free from the type of data augmentation assumption and data preprocessing; still, our method achieves better empirical performance than the data augmentation-based methods in many environments.

Further, we tested our method against the entropy regularization method, where the entropy of a policy is controlled by a coefficient weight of the policy entropy. Empirically, we observe that our method RPO performs better in most environments, and some choice of coefficient eventually leads to worse performance. Moreover, we observe that even when the agent is trained for a large simulation experience, the performance might not stay consistent. In particular, in IsaacGym environments, the agent has access to a large sample due to the high-performing simulation on GPU. This abundance of data may not necessarily improve or maintain the performance. In the classic CartPole environment, we observe that PPO agents quickly achieve a certain performance; however, it then fails to maintain it, and the performance quickly drops as we train for more. Similar results have been found in the OpenAI BipedalWalker environments. On the other hand, our method RPO shows robustness to more data and keeps improving or maintaining a similar performance as we keep training agents for more data.

In summary, we make the following contributions:

- We investigate whether keeping the policy stochastic throughout the training is beneficial for the RL training.
- We propose an algorithm, Robust Policy Optimization (RPO), that uses a perturbed Gaussian distribution to represent actions and is empirically shown to maintain a certain level of policy entropy throughout the training.
- We evaluate our method on 18 tasks from four RL benchmarks. Evaluation results show that our method RPO consistently performs better than standard PPO, two data argumentation-based methods RAD and DRAC, and entropy regularization.

## 2 PRELIMINARIES AND PROBLEM SETTINGS

**Markov Decision Process (MDP)**. An MDP is defined as a tuple $\mathcal{M} = (\mathcal{S}, \mathcal{A}, \mathcal{P}, \mathcal{R})$. Here an agent interacts with the environment and take an action $a_t \in \mathcal{A}$ at a discrete timestep $t$ when at state $s_t \in \mathcal{S}$. After that the environment moves to next state $s_{t+1} \in \mathcal{S}$ based on the dynamic transition probabilities $\mathcal{P}(s_{t+1}|s_t, a_t)$ and the agent recieves a reward $r_t$ according to the reward function $\mathcal{R}$.

**Reinforcement Learning** In a reinforcement learning framework, the agent interacts with the MDP, and the agent's goal is to learn a policy $\pi \in \Pi$ that maximizes cumulative reward, where $\Pi$ is the set of all possible policies. Given a state, the agent prescribes an action according to the policy $\pi$, and an optimal policy $\pi^* \in \Pi$ has the highest cumulative rewards.

**Policy Gradient**. The policy gradient method is a class of reinforcement learning algorithms where the objective is formulated to optimize the cumulative future return directly. In a deep reinforcement learning setup, the policy can be represented as a neural network that takes the state as input and output actions. The objective of this policy network is to optimize for cumulative rewards. In this paper, we focus on a particular policy gradient method, Proximal Policy Optimization (PPO) (Schulman et al., 2017), which is similar to Natural Policy Gradient (Kakade, 2001) and Trust Region Policy Optimization (TRPO) (Schulman et al., 2015). The following is the objective of the PPO:

$$\mathcal{L}_\pi = -\mathbb{E}_t\big[\frac{\pi_\theta(a_t|s_t)}{\pi_{\theta_{old}}(a_t|s_t)} A(s_t, a_t)\big] \tag{1}$$

, where $\pi_\theta(a_t|s_t)$ is the probability of choosing action $a_t$ given state $s_t$ at timestep $t$ using the current policy parameterized by $\theta$. On the other hand, the $\pi_{\theta_{old}}(a_t|s_t)$ refer to the probabilities of using an old policy parameterized by previous parameter values $\theta_{old}$. The advantage $A(s_t, a_t)$ is an estimation which is the advantage of taking action $a_t$ at $s_t$ compared to average return from that state.

In PPO, a surrogate objective is optimized by applying clipping on the current and old policy ratio on equation 1. In the case of the discrete action, the output is often used as the categorical distribution, where the dimension is the number of actions. On the other hand, in continuous action cases, the output can correspond to parameters of Gaussian distribution, where each action dimension is represented by mean and variance. Finally, the action is taken as the sample from this Gaussian distribution. In this paper, we mainly focus on the *continuous action* case and assume the Gaussian distribution for action.

**Entropy Measure** Entropy is a measure of uncertainty in the random variable. In the context of our discussion, we measure the entropy of a policy from the action distribution it produces given a state. The more entropic an action distribution is, the more stochastic the decision would be as the final action is eventually a sample from the output action distribution. This measure can be seen as the policy's stochasticity, and higher entropy distribution allows agents to explore the state space more. Thus, entropy can play a role in controlling the amount of exploration the agent performs during learning. In this paper, we focus on the entropy measure and refer to the stochasticity of policy by this measure.

# 3 ROBUST POLICY OPTIMIZATION (RPO)

Exploration at the start of training can help an agent to collect diverse and representative data. However, as time progresses, the agent might start to reduce the exploration and try to be more certain about a state. In standard PPO, we observe that the agent shows more randomness in action at the beginning (see Figure 1), and it becomes smaller as the training progresses. We measure this randomness in the form of entropy, which indicates the randomness of the actions taken. In the continuous case, a practical approach is to represent the action as a parameter of Gaussian distribution. The entropy of this distribution represents the amount of randomness in action, which also indicates how much the policy can explore new states.

However, in Gaussian distribution, the samples are concentrated toward the mean; thus, it can limit the exploration in some RL environments. This paper presents an alternative to this Gaussian distribution and proposes a new distribution that accounts for higher entropy than the standard Gaussian. The goal is to keep the entropy higher or at some levels throughout the agent training. In particular, we propose to combine Gaussian Distribution and Uniform distribution. At each timestep of the algorithm training, we perturb the mean $\mu$ of Gaussian $\mathcal{N}(\mu, \sigma)$ by adding a random number drawn from the Uniform $\mathcal{U}(-\alpha, \alpha)$ and get the perturbed mean $\mu'$. Then the sample for action is taken from the new distribution $\mathcal{N}(\mu', \sigma)$. In this setup, the Gaussian parameters still depend on the state, similar to the standard setup. However, the Uniform distribution does not depend on the states. Thus as training progresses, the standard Gaussian distribution can be less entropic as the policy optimizes to be more confident in particular states. However, the state-independent perturbation using the Uniform distribution still enables the policy to maintain stochasticity.

Figure 1 shows the diagram of the resulting Gaussian distributions. This diagram is generated by first sampling data from a Gaussian distribution with mean $\mu = 0.0$ and standard deviation $\sigma = 1.0$ and then corresponding perturbed mean $\mu' = \mu + z$ and standard deviation $\sigma = 1.0$, where $z \sim \mathcal{U}(-3.0, 3.0)$. We see that the values are less centered around the mean for our Perturb Gaussian than the standard. Therefore, samples from our proposed distribution should have more entropy than the standard Gaussian.

**Algorithms** 1 shows the details procedure of our RPO method. The agent first collects experience trajectory data using the current policy and stores it in a buffer $\mathcal{D}$ (lines 2 to 10). Next, the agent uses the standard Gaussian to sample the action in these interaction steps (line 6). Finally, these experience data are used to update the policy, optimizing for policy parameter $\theta$. In this case, given a state $s_t$, the policy network outputs $\mu$ and $\sigma$ for each action. Then the $\mu$ is perturbed by adding a value $z$, which samples from a Uniform distribution $\mathcal{U}(-\alpha, \alpha)$ (lines 13 and 14 in blue). After that, the probability is computed from the new distribution $\mathcal{N}(\mu', \sigma)$, which is eventually used to calculate action log probabilities that are used for computing loss $L_\pi$. In the case of the $\alpha$ hyperparameter, we report results using $\alpha = 0.5$ unless otherwise specified.

---

**Algorithm 1** Robust Policy Optimization (RPO)

---

1:  Initialize parameter vectors $\theta$ for policy network.
2:  **for** each iteration **do**
3:      $\mathcal{D} \leftarrow \{\}$
4:      **for** each environment step **do**
5:          $\mu, \sigma \leftarrow \pi_\theta(.|s_t)$
6:          $a_t \sim \mathcal{N}(\mu, \sigma)$
7:          $s_{t+1} \sim P(s_{t+1}|s_t, a_t)$
8:          $r_t \sim R(s_t, a_t)$
9:          $\mathcal{D} \leftarrow \mathcal{D} \cup \{(s_t, a_t, r_t, s_{t+1})\}$
10:     **end for**
11:     **for** each observation $s_t$ in $\mathcal{D}$ **do**
12:         $\mu, \sigma \leftarrow \pi_\theta(.|s_t)$
13:         $z \sim \mathcal{U}(-\alpha, \alpha)$
14:         $\mu' \leftarrow \mu + z$
15:         $prob \leftarrow \mathcal{N}(\mu', \sigma)$
16:         $logp \leftarrow prob(a_t)$
17:         Compute RL loss $L_\pi$ using $logp$, $a_t$, and value function.
18:     **end for**
19: **end for**

---

## 4    EXPERIMENTS

### 4.1    SETUP

**Environments** We conducted experiments on continuous control task from four reinforcement learning benchmarks: DeepMind Control (Tunyasuvunakool et al., 2020), OpenAI Gym (Brockman et al., 2016), PyBullet (Coumans & Bai, 2016–2021), and Nvidia IsaacGym (Makoviychuk et al., 2021). These benchmarks contain diverse environments with many different tasks, from low to high-dimensional environments (observations and actions space). Thus our evaluation contains a diverse set of tasks with various difficulties. The IsaacGym contains environments that run in GPU, thus enabling fast simulation, which eventually helps collect a large amount of simulation experience quickly, and faster RL training with GPU enables deep reinforcement learning models.

**Baselines** We compare our method RPO with the **PPO** (Schulman et al., 2017) algorithm. Here our method RPO uses the perturbed Gaussian distribution to represent the action output from the policy network, as described in Section 3. In contrast, the PPO uses standard Gaussian distribution to represent its action output. Further, we observe that the data augmentation method can help increase the policy's entropy by often randomly perturbing observations. This process might improve the performance where higher entropy is preferred. Thus, we compare our method with two data augmentation-based methods: **RAD** (Laskin et al., 2020), and **DRAC** (Raileanu et al., 2020). Here, The pure data augmentation baseline RAD uses data processing before passing it to the agent, and the DRAC uses data augmentation to regularize the value and policy network. Both of these data augmentation methods use PPO as their base RL policy. Another common approach to increase entropy is to use the **Entropy Regularization** in the RL objective. A coefficient determines how much weight the policy would give to the entropy. We observe that various weighting might result in different levels of entropy increment. We use the entropy coefficient 0.0, 0.01, 0.05, 0.5, 1.0, and 10.0 and compare their performance in entropy and, in return, with our algorithm's RPO. Note that our method does not introduce any additional hyper-parameters and does not use the entropy coefficient hyper-parameters. The implementation details of our algorithm and baselines are in the Appendix. To account for the stochasticity in the environment and policy, we run each experiment several times and report the mean and standard deviation. Unless otherwise specified, we run each experiment with **10 random seeds**.

### 4.2    RESULTS

We compare the return performance of our method with baselines and show their entropy to contrast the results with the entropy of the policy.

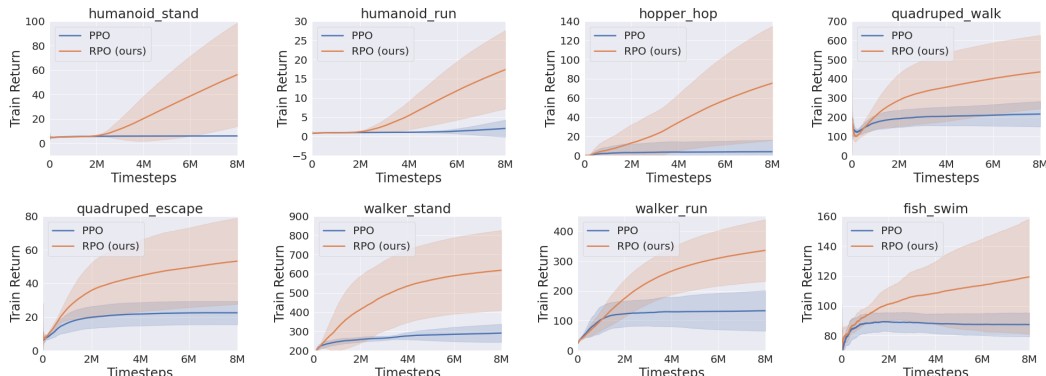

Figure 2: Results on DeepMind Control Environments. PPO agent fails to learn any useful behavior and thus results in low episodic return in some environments (humanoid stand, humanoid run, and hopper hop). Overall, our method RPO performs better and achieves a much higher episodic return. The RPO also shows a better mean return than the PPO in other environments. In many settings, PPO agents stop improving their performance after around 2M timestep while RPO consistently improves over the entire training time.

**Comparison with PPO** This evaluation is the direct form of comparison where no method uses any aid (such as data augmentation or entropy regularization) in the entropy value. Figure 2 shows results comparison on *DeepMind Control Environments*. Results on more environments are in Appendix Figure 11. The entropy comparison is given in the Appendix Figure 12. In most scenarios, our method RPO shows consistent performance improvement in these environments compared to the PPO. In some environments, such as humanoid stand, humanoid run, and hopper hop, the PPO agent fails to learn any useful behavior and thus results in low episodic return. In contrast, our method RPO shows better performance and achieves a much higher episodic return. The RPO also shows a better mean return than the PPO in other environments. In many settings, such as in quadruped (walk, run, and escape), walker (stand, walk, run), fish swim, acrobot swingup, the PPO agents stop improving the performance after around 2M timestep. In contrast, our agent RPO shows consistent improvement over time of the training. This performance gain might be due to the proper management of policy entropy, as in our setup, the agent is encouraged to keep exploring as the training progress. On the other hand, the PPO agent might settle in a sub-optimal performance as the policy entropy, in this case, decreases as the agent trains for more timesteps. These results show the effectiveness of our method in diverse control tasks with varying complexity.

Results of *OpenAI Gym* environments: Pendulum and BipedalWalker are in Figure 3. Overall, our method, RPO, performs better compared to the PPO. In Pendulum environments, the PPO agent fails to learn any useful behavior in this setup. In contrast, RPO consistently learns with the increase in timestep and eventually learns the task. We see the policy entropy of RPO increases initially and eventually remains at a certain threshold, which might help the policy to stay exploratory and collect more data. In contrast, the PPO policy entropy decreases over time, and thus eventually, the performance remains the same, and the policy stops learning. This scenario might contribute to the bad performance of the PPO.

In the BipedalWalker environment, we see that both PPO and RPO learn up to a certain reward quickly. However, as we keep training both policies, we observe that after a certain period, the PPO's performance drops and even starts to become worse. In contrast, the RPO stays robust as we train for more and eventually keep improving the performance. These results show the robustness of our method when ample train time is available. The entropy plot shows a similar pattern, as PPO decreases entropy over time, and RPO keeps the entropy at a certain threshold.

Figure 4 shows results comparison on *PyBullet* environments: Ant, and Minituar. We observe that our method RPO performs better than the PPO in the Ant environment. In the Minitaur environment, PPO quickly (at around 2M) learns up to a certain reward and remains on the same performance as time progresses. In contrast, RPO starts from a lower performance, eventually surpassing the PPO's performance as time progresses. These results show the robustness of consistently improving the

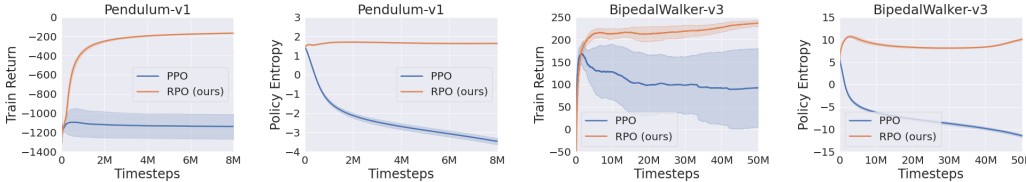

Figure 3: Results on OpenAI Gym Environments. The PPO agent fails to learn any useful behavior in the Pendulum environment. In contrast, RPO consistently learns with the increase in timestep and eventually achieves higher scores. The entropy of RPO policy increases initially and eventually remains at a certain threshold. In contrast, the PPO policy entropy decreases over time. In Bipedal-Walker, the PPO fails to progress after achieving a particular reward, ad eventually, the performance degrades with more training. In contrast, our RPO agent consistently improves performance over the entire training time.

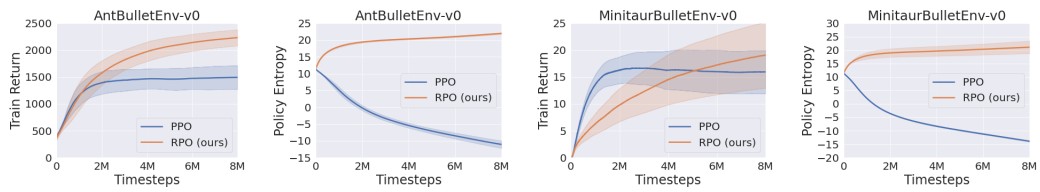

Figure 4: Results on PyBullet Environments. Our method RPO consistently performs better than the PPO in the Ant environment. On the other hand, in the Minitaur environment, PPO quickly learns up to a particular reward and remains on the same performance as time progresses while RPO surpasses the PPO's performance.

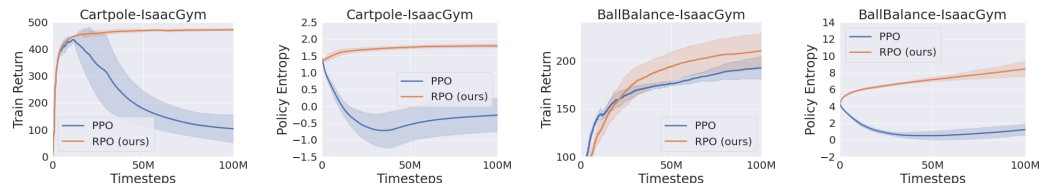

Figure 5: Results on IsaacGym Environments. In the Cartpole environment, the performance of PPO started to degrade over time, and the policy entropy kept decreasing. In contrast, our RPO agent keeps improving the performance over the entire training time. These results show the robustness of our method RPO over PPO even when an abundance of simulation is available. In the BallBalance environment, our method RPO achieves a slight performance improvement compared to PPO.

policy of the RPO method. The entropy pattern remains the same in both cases; PPO reduces entropy while RPO keeps the entropy at a certain threshold which it learns automatically in an environment.

Figure 5 shows results comparison on *IsaacGym* environments: Cartpole, and BallBalance. In this setup, we run the simulation up to 100M timesteps which take around 30 minutes for each run in each environment in a Quadro RTX 4000 GPU. We see that for Cartpole, both PPO and RPO learn the reward quickly, around $450$. However, as we kept training for a long, the performance of PPO started to degrade over time, and the policy entropy kept decreasing. On the other hand, our RPO agent keeps improving the performance; notably, the performance never degrades over time. The policy entropy shows that the entropy remains at a threshold. This exploratory nature of RPO's policy might help keep learning and get better rewards. These results show the robustness of our method RPO over PPO even when an abundance of simulation is available. Interestingly, more simulation data might not always be good for RL agents. In our setup, the PPO even suffers from further training in the Cartpole environment. In the BallBalance environment (results are averaged over 3 random seed runs), our method RPO achieves a slight performance improvement over PPO. Overall, our method RPO performs better than the PPO in the two IsaacGym environments.

Table 1: Result comparison with data augmentation on DeepMind control environments. Our method RPO performs better in mean episodic return in most environments than PPO and other data augmentation baselines RAD and DRAC. Moreover, in some environments, the data augmentation baselines worsen the performance compared to the base PPO. The results are after training the agent for 8M timesteps. The mean and standard deviations are over 10 seed runs.

| Env | PPO | RAD-PPO | DRAC-PPO | RPO (ours) |
|---|---|---|---|---|
| acrobot swingup | 23.93 ±6.0 | 13.41 ±9.84 | 21.32 ±13.17 | **40.46** ±4.01 |
| fish swim | 87.39 ±8.0 | 81.53 ±6.53 | 87.5 ±7.74 | **119.42** ±38.46 |
| humanoid stand | 6.06 ±0.17 | 6.14 ±0.37 | 37.58 ±66.09 | **55.22** ±35.34 |
| humanoid run | 1.55 ±1.59 | 1.1 ±0.04 | 13.13 ±16.6 | **13.26** ±9.45 |
| pendulum swingup | 518.65 ±279.73 | 23.04 ±17.08 | 320.51 ±319.76 | **699.79** ±32.97 |
| quadruped walk | 216.67 ±66.56 | 395.07 ±124.91 | 374.78 ±197.98 | **437.66** ±191.93 |
| quadruped run | 183.14 ±38.48 | 158.01 ±26.84 | 247.02 ±83.23 | **258.75** ±96.18 |
| quadruped escape | 22.51 ±7.0 | **54.0** ±22.13 | 52.62 ±21.89 | 53.37 ±25.73 |
| walker stand | 346.55 ±104.87 | 558.92 ±131.09 | 361.99 ±93.44 | **652.7** ±202.47 |
| walker walk | 329.61 ±67.92 | 333.22 ±84.38 | 425.51 ±150.52 | **611.88** ±170.46 |
| walker run | 123.63 ±53.5 | 141.43 ±34.77 | 132.28 ±37.18 | **291.83** ±108.8 |

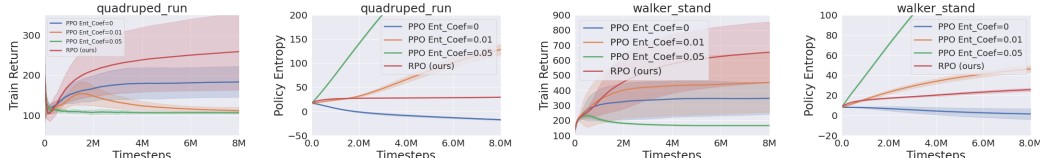

Figure 6: Comparison with Entropy Regularization. Sometimes (e.g., walker stand), the performance improves with a proper setup of tuned coefficient value. In other cases, the improper coefficient value often worsens the performance. In contrast, our method RPO consistently performs better than PPO and the different entropy coefficient baselines.

**Comparison with Data Augmentation**. In results on all DeepMind Control environments are shown in Table 1. Our method performs better in mean episodic return in most environments than PPO and other data augmentation baselines RAD and DRAC.

The return and policy entropy curve are shown in Appendix, Figure 10. We observe that the data augmentation slightly improved the base PPO algorithms, and the policy entropy shows higher than the base PPO. However, our method RPO still maintains a better mean return than all the baselines. The entropy of our method shows an increase at the initial timestep of the training. However, it eventually becomes stable at a particular value. The data augmentation method, especially RAD, shows an increase in entropy throughout the training process. However, this increase does not translate to the return performance. Moreover, data augmentation assumes some prior knowledge about the environment observation, and selecting a suitable data augmentation method requires domain knowledge. Improper handling of the data augmentation may result in worsen performance.

In addition, the augmentation method poses an additional processing time which can contribute to longer training time. In contrast, our method RPO does not require such domain knowledge and is thus readily applicable to any RL environment. Moreover, in return performance, our method shows better performance compared to the base PPO and these data augmentation methods.

**Comparison with Entropy Regularization**

Due to the variety in the environments, the entropy requirement might be different. Thus, improper setting of the entropy coefficient might result in bad training performance. In the tested environments, we observe e that sometimes the performance improves with a proper setup of tuned coefficient value and often worsens the performance (in Figure 6). However, our method RPO consistently performs better than PPO and the different entropy coefficient baselines. Importantly, our method does not use these coefficient hyper-parameters and still consistently performs better in various environments, which shows our method's robustness in various environment variability.

We observe that in some environments, the coefficient 0.01 improves the performance of standard PPO with coefficient 0.0 while increasing the entropy. However, an increase in entropy to 0.05 and

Figure 7: Ablation on $alpha$ values of the uniform distribution for RPO. An $\alpha$ value between 0.1 to 3 often results in better performance, while a large value often results in worse performance.

above results in an unbounded entropy increase. Thus, the performance worsens in most scenarios where the agent fails to achieve a reasonable return. Results with all the coefficient variants in the Appendix Figure 9. Overall, our method RPO achieves better performance in the evaluated environments in our setup. Moreover, our method does not use the entropy coefficient hyperparameter and controls the entropy level automatically in each environment.

**Ablation Study** We conducted experiments on the $\alpha$ value ranges in the Uniform distribution. Figure 7 shows the return and policy comparison. We observe that the value of $\alpha$ affects the policy entropy and, thus, return performance. A smaller value of $\alpha$ (e.g., 0.001) seems to behave similarly to PPO, where policy entropy decreases over time, thus hampered performance. Higher entropy values, such as 1000.0, make the policy somewhat random as the uniform distribution dominates over the Gaussian distribution (as in Algorithm 1. This scenario keeps the entropy somewhat at a constant level; thus, the performance is hampered. Overall, a value between 0.1 to 3 often results in better performance. Due to overall performance advantage, in this paper, we report results with $\alpha = 0.5$ for all 18 environments. Learning curve for more environments are in Appendix Figure 13.

## 5 RELATED WORK

Since the inception of the practical policy optimization method PPO (TRPO and Natural Policy Gradient), several studies have investigated different algorithmic components (Engstrom et al., 2019; Andrychowicz et al., 2020; Ahmed et al., 2019; Ilyas et al., 2019). Entropy regularization enjoys faster convergence and improves policy optimization (Williams & Peng, 1991; Mei et al., 2020; Mnih et al., 2016; Ahmed et al., 2019). However, some empirical findings observe the difficulty in setting proper entropy setup and observe no performance gain in many environments (Andrychowicz et al., 2020). Gaussian distribution is commonly used to represent continuous action (Schulman et al., 2015; 2017). Thus many follow-ups to implementation also use this Gaussian distribution as s default setup (Huang et al., 2022; 2021). In contrast, in this paper, we propose perturbed Gaussian distribution to represent the continuous action to maintain policy entropy. Furthermore, in this paper, we showed the empirical advantage of our method in several benchmark environments compared to the standard Gaussian-based method. Due to space limitations, we moved some related work discussions to the Appendix.

## 6 CONCLUSION

This paper investigates the effect of entropy in policy optimization-based reinforcement learning. We design a method that, when applied to a policy network, can keep the entropy of the policy to a certain threshold through training. Our approach uses a perturbed Gaussian distribution by a random number sample from a uniform distribution to represent the policy. We observe that keeping a certain entropy threshold throughout using our method can help better policy learning. Our proposed algorithm Robust Policy Optimization (RPO), performs better than the standard PPO on 18 continuous control tasks from four RL benchmarks, DeepMind Control, OpenAI Gym, Pybullet, and IsaacGym. Further, our method performs better in many settings than other approaches, encouraging higher entropy, such as data augmentation and entropy regularization. Overall, we show our method's effectiveness and robustness where standard PPO or other baseline methods fail to maintain or even worsen the performance over time.

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

# A  APPENDIX

## A.1  ADDITIONAL RELATED WORK

Another form of approach which adds entropy to the policy is data augmentation. We observe that the policy entropy often increases when the data is augmented often by some form of random processes. This improvement in entropy can eventually lead to better exploration in some environments. However, the type of data augmentation might be environment specific and thus requires domain knowledge as to what kind of data augmentation can be applied. In contrast, to this method, our method maintains an entropy threshold during entire policy learning. Moreover, we evaluated our algorithm with two kinds of data augmentation-based methods RAD (Laskin et al., 2020) and DRAC (Raileanu et al., 2020). Finally, we evaluate with an effective data augmentation approach random amplitude modulation, which was found to be worked well in vector-based states set up.

Many other forms of improvement have been investigated in policy optimization, which results in better empirical success, such as Generalized Advantage Estimation (Schulman et al., 2016), Normalization of Advantages (Andrychowicz et al., 2020), and clipped policy and value objective (Schulman et al., 2017; Engstrom et al., 2019; Andrychowicz et al., 2020). These improvements eventually led to overall improvement and are now used in many standard implementations such as (Huang et al., 2022; 2021). In our implementation, we leverage these implementation tricks when possible. Thus our method works on top of these essential improvements. Furthermore, our method is complementary to these approaches and thus can be used along with this method. In this paper, our implementation of baselines also used these implementation tricks, which is for a fair comparison.

## A.2  EXPERIMENTS

**Implementation Details**: Our algorithm and the baselines are based on the PPO (Schulman et al., 2017) implementation available in (Huang et al., 2022; 2021). This implementation incorporated many important advancements from existing literature in recent years on policy gradient (e.g., Orthogonal Initialization, GAE, Entropy Regularization). We refer reader to (Huang et al., 2022) for further references. The pure data augmentation baseline RAD (Laskin et al., 2020) uses data processing before passing it to the agent, and the DRAC (Raileanu et al., 2020) uses data augmentation to regularize the loss of value and policy network. We experimented with vector-based states and used *random amplitude scaling* proposed in RAD (Laskin et al., 2020) as a data augmentation method for RAD and DRAC. In the random amplitude scaling, the state values are multiplied with random values generated uniformly between a range $\alpha$ to $\beta$. We used the suggested (Laskin et al., 2020) and better performing range $\alpha = 0.6$ to $\beta = 1.2$ for all the experiments. Moreover, both RAD and DRAC use PPO as their base algorithm. However, our RPO method does not use any form of data augmentation.

Table 2: Hyperparameters for the experiments.

| Description | Value |
|---|---|
| Number of rollout steps | 2048 |
| Learning rate | $3e-4$ |
| Discount factor gamma | 0.99 |
| Lambda for the GAE | 0.95 |
| Number of mini-batches | 32 |
| Epochs to update the policy | 10 |
| Advantages normalization | True |
| Surrogate clipping coefficient | 0.2 |
| Clip value loss | Yes |
| Value loss coefficient | 0.5 |

We use the hyperparameters reported in the PPO implementation of continuous action spaces (Huang et al., 2022; 2021), which incorporate best practices in the continuous control task. Furthermore, to mitigate the effect of hyperparameters choice, we keep them the same for all the environments. Further, we keep the same hyperparameters for all agents for a fair comparison. The common hyperparameters can be found in Table 2.

**Entropy Regularization** The results comparison of RPO with entropy coefficient are in Figure 8

**Data augmentation return curve and entropy**: Figure 10 shows data augmentation return curve and entropy comparison with RPO.

**Results on DeepMind Control** is in Figure 11.

**Entropy Plot for DeepMind Control** is in Figure 12.

**Ablation Study** Ablation on $\alpha$ values of RPO are shown in Figure 13.

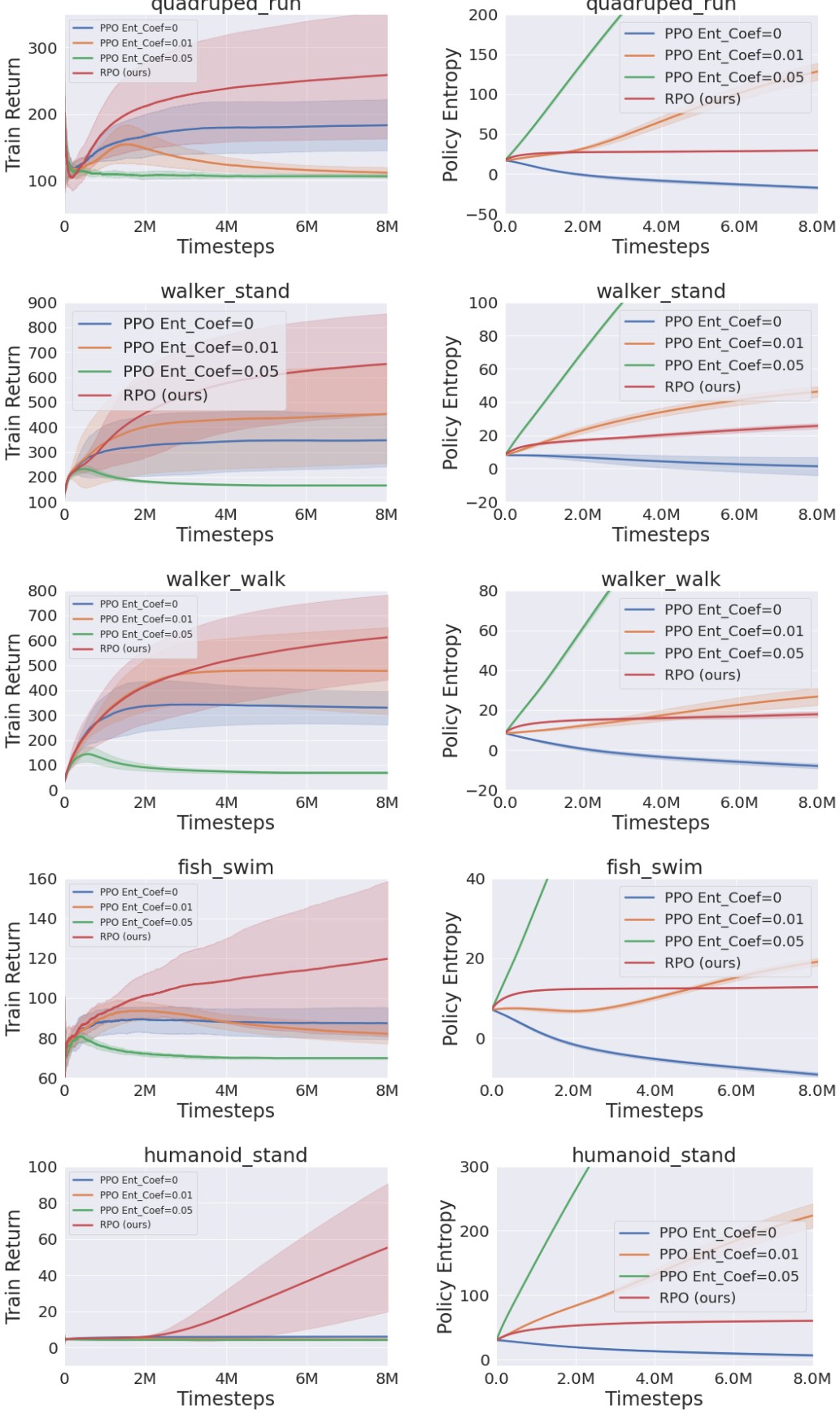

Figure 8: Comparison with Entropy Regularization.

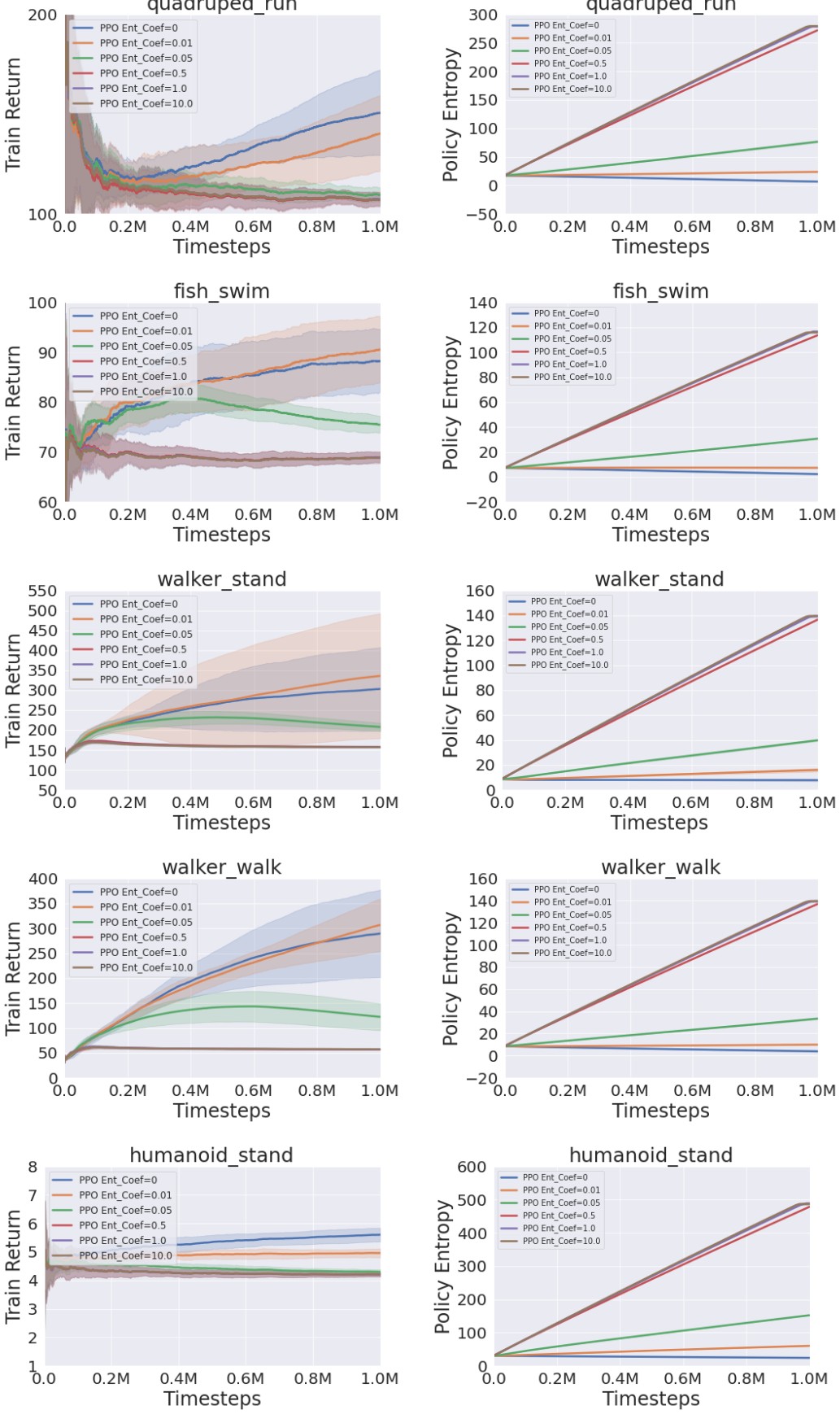

Figure 9: Comparison with Entropy Regularization on different coefficient values.

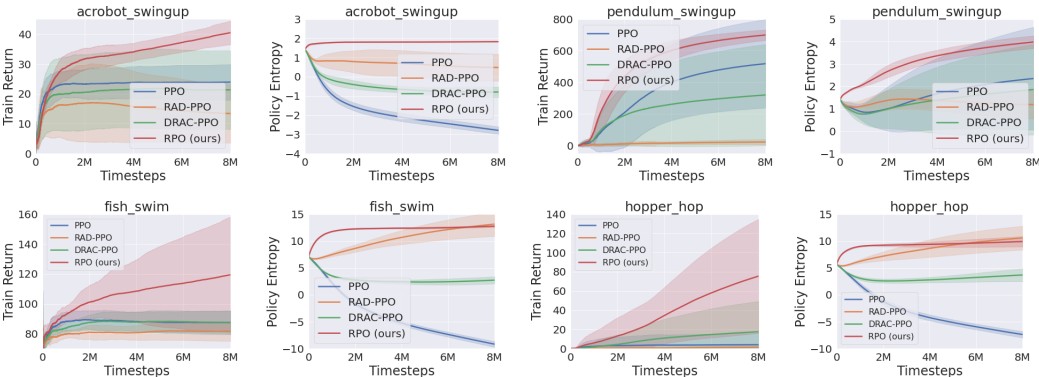

Figure 10: Comparison with PPO and data augmentation RAD, and DRAC.

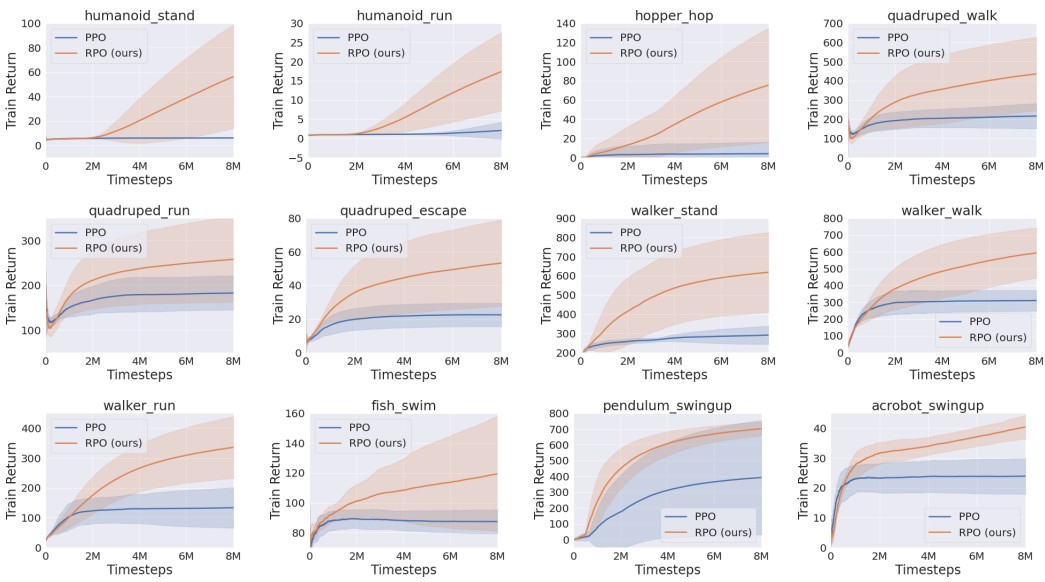

Figure 11: Results on DeepMind Control Environments. PPO agent fails to learn any useful behavior and thus results in low episodic return in some environments (humanoid stand, humanoid run, and hopper hop). Overall, our method RPO performs better and achieves a much higher episodic return. The RPO also shows a better mean return than the PPO in other environments. In many settings, PPO agents stop improving their performance after around 2M timestep while RPO consistently improves over the entire training time.

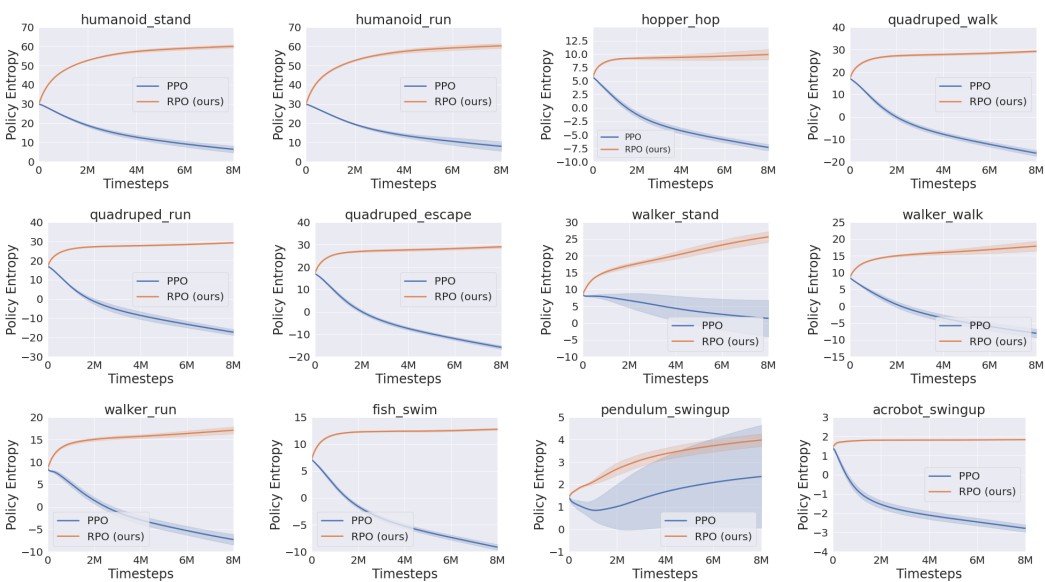

Figure 12: Results on DeepMind Control Environments.

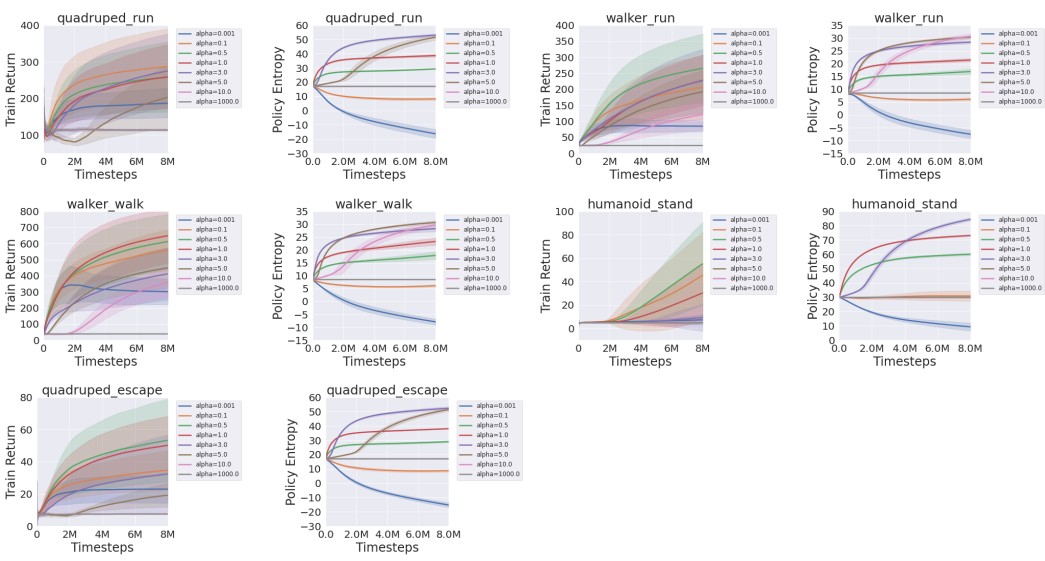

Figure 13: Ablation on $alpha$ values of the uniform distribution for RPO.

