# OpenReview forum: "Robust Policy Optimization in Deep Reinforcement Learning"
_ICLR.cc/2023/Conference — Submitted to ICLR 2023_

### Official Review · Reviewer_fgod · 2022-10-24

**Confidence:** 3
**Correctness:** 2
**Technical Novelty And Significance:** 1
**Empirical Novelty And Significance:** 1
**Recommendation:** 3

**Clarity, Quality, Novelty And Reproducibility:**

Clarity/Quality/Novelty: See questions under “Strengths and Weaknesses.”

Reproducibility: The paper includes pseudo-code and hyperparameter values used to implement their algorithm. I believe a reader could re-implement the proposed modification from these details.

**Strength And Weaknesses:**

The paper is overall written well. However, it’s not clear to me how adding action noise is different from existing ways of incorporating entropy regularization. For example, this is equivalent to enforcing that each component of \sigma is at least some threshold. This clipping already exists in many PPO implementations I believe.

I also have some questions and concerns in regards to the experimental evaluation:
- Figure 2 shows that PPO fails to learn in these environments, but this is probably due to the entropy regularization being removed. A fairer comparison would be to PPO with the standard entropy regularization.
- The comparison to the data augmentation methods is a bit confusing to me since these are methods that perform image augmentation for image-based RL tasks. Are the experimental tasks being solved from images?
- There seems to be more useful comparisons to include, such as Soft Actor-Critic, which optimizes the maximum-entropy RL objective and automatically tunes the entropy coefficient, and exploration strategies for RL, such as Random Network Distillation.
- What are the action spaces for each environment? It is my understanding that each action component is between -1 and 1 for the DM Control Suite tasks. Wouldn’t alpha values larger than or equal to 1 essentially make the actions uniform-random?

**Summary Of The Paper:**

This paper argues for stochastic policies that maintain high entropy for high RL performance. In particular, the authors propose to randomly perturb the mean of the output Gaussian distribution to produce more diverse actions. In the experimental evaluation, this perturbation scheme is compared to PPO (with and without entropy regularization) and data augmentation techniques including RAD and DRAC.

**Summary Of The Review:**

Overall, the novelty of the work is not clear to me. The experimental evaluation also lacks relevant baselines, and some details of their evaluation are currently unclear. Therefore, I am recommending a reject.

---

> ### Author Response · Authors · 2022-11-14
> **Response to reviewer fgod**
>
> We compare with entropy regularization as one of our baselines (please refer to “Comparison with Entropy Regularization” in the result section 4.2). With the entropy regularizer, the PPO still fails (Figure 6) in many scenarios. For details results on varying the co-efficient values, please refer to Figure 9 in Appendix.
>
> We observe that incorporating entropy regularization results in a rather linear increase in policy entropy (Figures 6 and 9). Thus, it might not be beneficial in general; however, depending on the task, it might improve the performance of the base algorithm. We observe a performance gain with an entropy co-efficient of 0.01 in the walker stand environment (Figure 6, 3rd plot); however, our method RPO achieves better results than all co-efficient values tested.
>
> We choose to evaluate the data augmentation method as they often use randomness in augmenting observation. We observe this process shows to increase policy entropy. Further, we compare our method with entropy regularization, which directly enforces higher entropy in the objective function.
>
> This paper considers vector-based states (robot joints, sensors), and data augmentation (i.e., random amplitude modulation)  is applied in this state. This approach was proposed in prior work (RAD), which was shown to improve performance.
>
> We primarily investigate the policy gradient-based algorithm (e.g., on-policy PPO). However, incorporating and comparing with off-policy SAC and other exploratory strategies would be interesting for future work.
>
> Action space is continuous and represented as Gaussian distribution. In the implementation, we ensure the output action is within the action boundary of the environment; we clip the action if it is outside the environmental action space. The alpha value perturbs the mean of the distribution. However, the sample from this distribution can still be within the limit of a given action space.

---

> > ### Comment · Reviewer_fgod · 2022-11-18
> > **Response to authors**
> >
> > Thanks for the clarifications! It would be good to mention that the comparison is specifically to the random amplitude modulation of RAD when first referencing RAD in the paper. Since RAD is often known for its use of image augmentations, this can be confusing to some readers.

---

### Official Review · Reviewer_NT5y · 2022-10-24

**Confidence:** 3
**Correctness:** 3
**Technical Novelty And Significance:** 2
**Empirical Novelty And Significance:** 2
**Recommendation:** 3

**Clarity, Quality, Novelty And Reproducibility:**

- The paper is not clear at points (for ex: Algo. 1)
- The novelty is limited in the paper.
- The method seems easy to reproduce.

**Strength And Weaknesses:**

Some strengths of the paper:

- The idea is easy to understand, and extremely easy to implement. Basically just adding a random number from uniform distribution would work.
- The results show that the agent trained with the proposed method explores more than the baseline PPO method.

Some weaknesses of the paper:

- The idea of exploration-exploitation is very old in the RL/bandit community.
- Authors didn't compare the std. exploration methods in the literature.
- There is no theoretical justification for choosing this particular distribution. How do other distributions with wider support work in place of gaussian?
- The effect of changing the distribution to the constraint of the PPO method is also not discussed. How does the constraint function behave when the policy parameterization is changed?
- Algorithm 1 is not written very well. what is "logp", "prob"? Please make it more clear.

**Summary Of The Paper:**

The authors propose a Robust Policy Optimization (RPO) method, a simple extension of the popular Proximal Policy Optimization (PPO) method, by adding a uniform random number to the action mean, sampled from a normal distribution. It is argued that std. normal distribution used to parameterize continuous action is not suitable if more exploration is expected from the agent. Authors modify the gaussian to have more support than the std. gaussian r.v.

Empirical results on std. RL benchmarks demonstrate that the proposed method has higher exploration throughout the training.

**Summary Of The Review:**

- Authors propose a modification of the existing PPO method for continuous actions cases, by increasing the support of the policy parameterization distribution.
- They add a uniform random number to the mean of the gaussian distribution used to parameterize the policy, thereby increasing its support.
- There is no theoretical justification for using the particular distribution; related works on using std. exploration ($\eps$-greedy etc) is not discussed.
- Other distributions with wider support around the mean are also not discussed in the paper.

---

> ### Author Response · Authors · 2022-11-14
> **Response to reviewer NT5y**
>
> We choose to evaluate the data augmentation method as they often use randomness in augmenting observation. We observe this process shows to increase policy entropy. Further, we compare our method with entropy regularization, which directly enforces higher entropy in the objective function.
>
> In this paper, we evaluate the effectiveness of our approach to Gaussian policy only, which is commonly used in many existing algorithms for continuous control tasks. However, applying uniform randomness to action distribution is a general approach and might be integrated into other (non-Gaussian) scenarios.
>
> $prob$ is the probability of all actions given an observation $s_t$. Then the $logp$ is the $\log$ of the probability of action $a_t$.

---

> > ### Comment · Reviewer_NT5y · 2022-11-16
> > **Response to authors comments**
> >
> > Thank you authors for your response, and thanks for the clarification.

---

### Official Review · Reviewer_JmVS · 2022-10-25

**Confidence:** 3
**Correctness:** 3
**Technical Novelty And Significance:** 3
**Empirical Novelty And Significance:** 2
**Recommendation:** 3

**Clarity, Quality, Novelty And Reproducibility:**

The paper is generally clearly written with most concepts well explained, though some parts of this paper will be benefited by more explanations (especially the motivations/intuitions of the main algorithm). This work also detailed the experimental setup and algorithm settings (hyper-parameters). It also includes links to pseudo-code for users to reproduce their results.

However, my main concern about this work is its novelty. Perturbing a policy to achieve better exploration (entropy) is not a new idea and more comparisons and theoretical analysis are needed to back the authors' claims.

**Strength And Weaknesses:**

Strengths:
Paper is generally clear in explaining the ideas of  how a perturbed Gaussian policy policy can maintain policy entropy throughout training and help with RL performance.
The paper includes pseudo-code and detailed hyper-parameters for readers to re-implement their work. This work should be reproducible.
The authors evaluated the RPO method on quite a number of domains and in some cases the proposed method is more powerful than SOTA.

Weaknesses:
The justification of this method is mainly empirical, there are not theoretical justifications of this work in terms of why the proposed method would work.
If the main contribution is on increasing entropy for better exploration, then there is not enough baseline comparisons with other RL methods that perturb the policy to improve robustness as well as performance during training. For example, one such algorithm is NoisyNet, which also perturbs Gaussian policy network to improve robustness.



**Summary Of The Paper:**

In this work the authors proposed a new robust policy optimization method in which the main approach is to randomly perturb the mean of the Gaussian policy to improve exploration. The main motivation is that stochastic policies that have high entropy tend to produce better RL performance, and so the authors propose to constantly perturb this policy during training. Empirically they showed that this new policy outperforms standard RL methods including the ones with data augmentation schemes (e.g., PPO, RAD, DRAC).


**Summary Of The Review:**

Authors propose a new randomization scheme for RL policy training in order to improve exploration, and thus the performance of RL (PPO).
While this method may show some empirical benefits in some small domain, there has not been much theoretical backing of this approach justified in the paper. Furthermore, it is unclear whether this method will also work with other setting in which a non-Gaussian policy is used. Finally, the experiments are not fully convincing (for example in some cases the proposed method does not strongly beat the baseline, and more SOTA baselines are needed to really showcase the potential superiority of the proposed method).

---

> ### Author Response · Authors · 2022-11-14
> **Response to reviewer JmVS**
>
> We choose to evaluate the data augmentation method as they often use randomness in augmenting observation. We observe this process shows to increase policy entropy. Further, we compare our method with entropy regularization, which directly enforces higher entropy in the objective function.
> The NoisyNet idea is interesting and demonstrates a way to incorporate stochasticity in policy learning. However, unlike NoisyNet, our method does not require learning the noise parameters; instead, we use a simple uniform random number generator to incorporate stochasticity in policy learning.
>
> In this paper, we evaluate the effectiveness of our approach to Gaussian policy only, which is commonly used in many existing algorithms for continuous control tasks. However, applying uniform randomness to action distribution is a general approach and might be integrated into other (non-Gaussian) scenarios.

---

### Official Review · Reviewer_JnzH · 2022-10-31

**Confidence:** 4
**Correctness:** 2
**Technical Novelty And Significance:** 2
**Empirical Novelty And Significance:** 2
**Recommendation:** 3

**Clarity, Quality, Novelty And Reproducibility:**

Clear to read, I do not see any major issues with reproducibility. The proposed method appears to be novel.

**Strength And Weaknesses:**

Strength:
- Proposed method is extremely straightforward, easy to implement, and appears to show good results.
- Paper was easy to read and clearly written.

Weakness:
-  My initial problem with the paper is the word “robust” since the term itself is quite overloaded and can have different meanings in different fields (e.g. robust control has a very specific meaning). The authors’ use of the robustness throughout the paper seem quite vague and note well defined.
- While the experiments are quite detailed and do show improvements on a set of challenging tasks over the benchmark, I do not believe they provided us with a better understanding of why the proposed approach worked well. Especially since the idea that its beneficial to maintain a certain degree of high entropy through training seems somewhat counter-intuitive in many scenarios. I believe some toy examples would be more helpful in this regard in showing us why/under what conditions the proposed method works well, I think this is something that would greatly enhance the quality of the paper.


**Summary Of The Paper:**

This paper introduces a new algorithm RPO which builds on top of PPO. The algorithm takes the mean output of a Gaussian distribution, adds a uniform perturbation to the mean, then samples actions from the resulting Gaussian distribution with the perturbed mean. The proposed method is shown to be effective on a variety of high dimensional continuous control tasks.

**Summary Of The Review:**

Overall, while the introduced method does appear to be novel and shows evidence of empirical improvements on a set of benchmarks, I don’t believe the paper in its current state advances our understanding of the field as a whole and therefore I cannot recommend the paper for acceptance at this time.

---

> ### Author Response · Authors · 2022-11-14
> **Response to reviewer JnzH**
>
> We agree that the term ‘robust’ is indeed overloaded. Here, we use the term as the ability of the agent to be exploratory throughout the training, which can prevent (as demonstrated empirically) early convergence to bad policy (Pendulum-v1, Hoppep, Humanoid) or collapse from a good performing policy in the later stage of training (IsaacGym Cartpole, BipedalWalker-v3).
>
> We observe that our method of applying randomness to the action helps when the current method fails to perform due to convergence to a bad policy early in training, which may be due to a lack of exploration.

---

### Author Response · Authors · 2022-11-14
**To all reviewers**

Thanks to all reviewers for their insightful comments.
We are glad that the reviewers find the proposed method novel [JnzH], easy to understand, easy to implement, and reproducible [JnzH, NT5y, JmVS, fgod].

This paper is centered around observing the effect of maintaining a certain level of entropy throughout the training, thus encouraging exploration. This process is counter-intuitive (as pointed out by reviewer JnzH) to the existing understanding of policy learning, where the norm is to reduce entropy as the training progresses. We give an instantiation of the proposed idea and develop a practical algorithm RPO. We provide empirical evidence of the benefit of maintaining entropy in some challenging benchmark environments. We hope our work and empirical findings will encourage further investigation into this concept, especially from a theoretical perspective.

---

### Decision · Program_Chairs · 2023-01-20

**Decision:**

Reject

**Justification For Why Not Higher Score:**

The paper has significant deficiencies that all reviewers agree on.

**Justification For Why Not Lower Score:**

N/A

**Metareview: Summary, Strengths And Weaknesses:**


The paper presents a new algorithm called Robust Policy Optimization (RPO) for improving policy learning in reinforcement learning. RPO uses a perturbed Gaussian distribution to encourage high-entropy actions and maintain a certain entropy threshold throughout training. The paper demonstrates the effectiveness of RPO on a variety of continuous control tasks, showing that it outperforms other methods, such as data augmentation and entropy regularization.

The reviewers appreciated the simplicity of the proposed method. However, the reviewers showed significant concerns that lowered their confidence in the work. We encourage making the following improvements and submitting the paper to the next venue:
- Using a toy environment such as grid-world to clearly show that maintaining the entropy is more beneficial than diminishing the entropy
- Using the proposed approach on top of other policy gradient methods such as soft actor critic.
- Giving better justification for using perturbed Gaussian distribution. For example, the idea needs to be connected with related ideas. The proposed idea is equivalent to having a minimum threshold for standard deviation, which is crudely enforcing entropy regularization. It is not clear why the proposed idea would perform better than entropy regularization. A discussion and analysis relating to these ideas are warranted.